# Chitosan-olive oil microparticles for phenylethyl isothiocyanate delivery: Optimal formulation

**Ezequiel R. Coscueta** [1]*, **Ana Sofia Sousa**[1], **Celso A. Reis** [2,3,4], **Manuela Pintado**[1]*

1 Universidade Católica Portuguesa, CBQF—Centro de Biotecnologia e Química Fina–Laboratório Associado, Escola Superior de Biotecnologia, Porto, Portugal, 2 i3S - Instituto de Investigação e Inovação em Saúde, Universidade do Porto, Porto, Portugal, 3 Institute of Molecular Pathology and Immunology of University of Porto, Ipatimup, Porto, Portugal, 4 Medical Faculty, University of Porto, Al. Prof. Hernâni Monteiro, Porto, Portugal

* ecoscueta@porto.ucp.pt (ERC); mpintado@porto.ucp.pt (MP)

**Data Availability Statement:** All relevant data are within the manuscript and its Supporting Information files.

## Abstract

Phenylethyl isothiocyanate (PEITC), a chemopreventive compound, is highly reactive due to its considerably electrophilic nature. Furthermore, it is hydrophobic and has low stability, bioavailability and bioaccessibility. This restricts its use in biomedical and nutraceutical or food applications. Thus, the encapsulation of this agent has the function of overcoming these limitations, promoting its solubility in water, and stabilizing it, preserving its bioactivity. So, polymeric microparticles were developed using chitosan-olive oil-PEITC systems. For this, an optimisation process (factors: olive oil: chitosan ratio and PEITC: chitosan ratio) was implemented through a 3-level factorial experimental design. The responses were: the particle size, zeta-potential, polydisperse index, and entrapment efficiency. The optimal formulation was further characterised by FTIR and biocompatibility in Caco-2 cells. Optimal conditions were olive oil: chitosan and PEITC: chitosan ratios of 1.46 and 0.25, respectively. These microparticles had a size of 629 nm, a zeta-potential of 32.3 mV, a polydispersity index of 0.329, and entrapment efficiency of 98.49%. We found that the inclusion process affected the optical behaviour of the PEITC, as well as the microparticles themselves and their interaction with the medium. Furthermore, the microparticles did not show cytotoxicity within the therapeutic values of PEITC. Thus, PEITC was microencapsulated with characteristics suitable for potential biomedical, nutraceutical and food applications.

## 1. Introduction

Phenylethyl isothiocyanate (PEITC) is released from the enzymatic hydrolysis of gluconasturtiin, the most abundant glucosinolate found in watercress (a vegetable from the family *Brassicaceae*) by the enzyme myrosinase [1]. Among all the isothiocyanates (ITCs), PEITC is one of the most extensively studied with various biological activities such as antimicrobial [2], antioxidant and anti-inflammatory [3,4]. Several studies suggested that PEITC exhibits cancer preventive and therapeutic effects on multiple types of cancers [1,5–7] and is one of the ITCs in

**Funding:** This research was funded by Foundation for Science and Technology (FCT) and for Competitiveness and Internationalization Operational Program through the project n° 032094"GastroCure - Bioactive Soybean and Cruciferous extracts towards application in gastrointestinal disorders: development, characterization and delivery".

**Competing interests:** The authors have declared that no competing interests exist.

current clinical trials [8]. Although PEITC is biologically useful and has excellent potential as a health-promoting compound, its industrialisation has been limited due to its relative volatility and instability [9]. PEITC is a highly reactive electrophile, which is susceptible to the attack by nucleophilic molecules [4]. Furthermore, PEITC is a compound with low molecular weight (MW = 163.2 g/mol) and considerable hydrophobicity (logP = 3.47). Its pharmacokinetic feature includes first-order linear absorption with a high protein binding nature [10]. Therefore, its stabilisation becomes a technological challenge. Cyclodextrin was reported as a plausible carrier for ITCs [11]. Besides, PEITC was already stabilised with vegetable oils, such as olive oil, once vegetable oil protects non-polar ITCs from decomposition or volatilisation [9].

Currently, methods for bioactive compounds administration are increasingly sought in a selective, non-invasive way and with minimal side effects. Hence, microencapsulation is a technology used to improve the delivery strategy, promoting the controlled release and protect bioactive compounds, preserving their properties from undesirable reactions, while improving their functionality and bioavailability [12,13]. Additionally, polymeric microparticles (MPs) gained significant importance as they are biodegradable, biocompatible and can be formulated using a sustainable approach [14]. In that sense, chitosan (CS) is biocompatible, biodegradable, and non-toxic, making it an ideal candidate as a biomaterial for biomedical applications, namely drug delivery systems [15,16]. CS is also mucoadhesive since it can establish connections between the charged groups from mucins. This property gives CS better selectivity at mucosal injured sites, where there is usually a higher expression of mucins. This allows for increasing its concentration at the site of absorption [17]. CS was also recognised by the United States Food and Drug Administration (US FDA) as Generally Recognised as Safe (GRAS), so its consumption is safe [18].

Microencapsulation is a strategy that has rarely been addressed as a way of administering PEITC or any ITCs [12]. Dharmala et al. developed a MP using stearic acid and chitosan to deliver PEITC, based on Solid Lipid Nanoparticle methodology. Their study demonstrated that Chitosan–SLN MPs loaded with PEITC significantly improved the chemotherapeutic efficacy of PEITC [19]. These claims showed PEITC can be stabilised by microencapsulation, a technique sparingly explored for ITCs so far. According to the few studies of microencapsulated ITCs, some methods seem more promising, mainly ionic gelation and complex coacervation. Both methods used chitosan, although so far they have not been explored with PEITC [12].

In this context, in which we established that PEITC could be stabilised, on the one hand, by liquid lipids such as olive oil, and on the other hand, by microencapsulation, what would result from the combination of both? Currently, there is no study about the encapsulation of PEITC (or any ITCs), using chitosan and a liquid lipid, which opens a relevant research gap. This may lead to new MPs as delivery systems for bioactive compounds with potential use in biomedical and nutraceutical fields.

This work is part of a project that seeks to apply PEITC to treat ailments in the gastrointestinal tract (Fig 1). For this, it is necessary to design MPs for oral administration capable of releasing their content in response to a given stimulus. Specifically, the MPs to be developed will be destined to the colon (Fig 1), so during their course, they must face a series of adverse conditions that result from the digestion process, which are detrimental to PEITC in its free state. So, this work aims to use CS as a polymer for the encapsulation of PEITC. On the other hand, it is also intended to give an even more innovative approach by combining CS and a commonly used liquid lipid, the nutritious and healthy olive oil (Fig 1). Thus, not only will the encapsulation system be explored, but it will also be optimised through a 3-level factorial experimental design. Finally, this work intends to select green techniques in the preparation of MPs, minimising the environmental impact and being as innocuous as possible for potential human use.

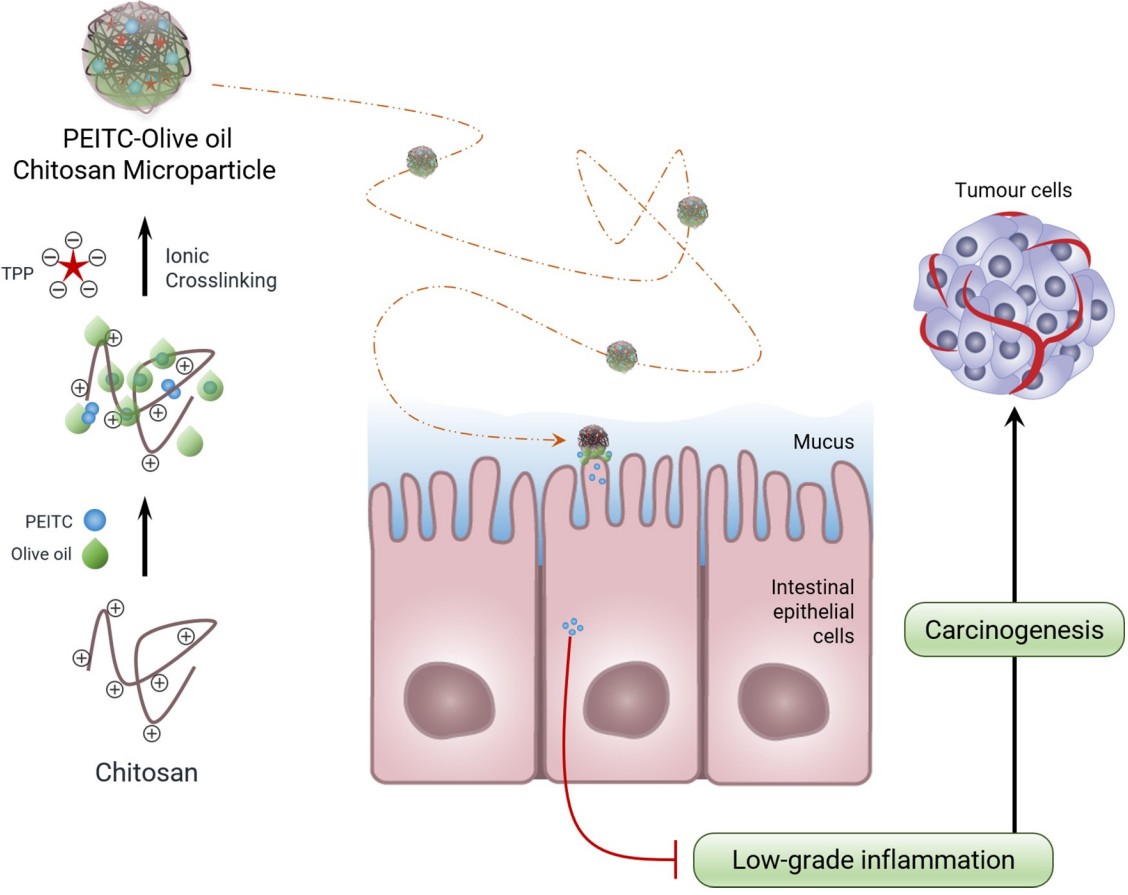

**Fig 1. Overview of the great challenge.** Representation of the global hypothesis that guides the specific objective of this initial development work.

## 2. Materials and methods

### 2.1. Chemicals

Low molecular weight chitosan (CS) was obtained from Sigma-Aldrich (St. Louis, USA) and possessed a deacetylation degree between 75 and 85% and a molecular weight of 107 kDa. Sodium Tripolyphosphate (TPP) and MEM non-essential amino acid solution were supplied by Sigma-Aldrich (Merck, Darmstadt, Germany). Dulbecco's Modified Eagle Medium (DMEM) high glucose and Penicillin-Streptomycin mixture were obtained from Lonza (Basel, Switzerland). Fetal bovine serum (FBS) was purchased from Biowest (Nuaillé, France). All the reagents were used as received without further purification. PEITC standard was supplied by Santa Cruz Biotechnology Inc. (Dallas, Texas, USA). The olive oil was an extra virgin, purchased in a local store. All the other reagents were of analytical grade and used without further purification. Ultrapure water was used to prepare all the solutions.

### 2.2. Microparticle production

MPs were produced using the ionic gelation methodology through adaptation of the procedure described by Madureira, Pereira, Castro and Pintado [20]. Briefly, CS was dissolved at 5 mg/mL in acetic acid 1%, dissolved for 72 h stirred at room temperature. For later use, the pH value was adjusted to 5.0 with NaOH. TPP was used at CS to TPP relation of 7:1. As MPs were

produced, void ultra-pure water was used to preserve the working volumes previously described. For all experiments the CS concentration was constant. In the case of PEITC-loaded CS MPs, the process began with the addition of 4 mL of CS and an amount of PEITC to maintain the PEITC: CS mass ratio of 0.5 to 1.0. The system was filled to 9 mL with ultra-pure water, which was then placed under gentle stirring and 1 mL of TPP were added, dropwise with a syringe pump, at room temperature. In the case of mixed PMs of CS and olive oil loaded with PEITC, the process was the same, but with the addition of olive oil before the addition of PEITC. In the case of optimisation experiments, the process remained the same but varying the proportions of the PEITC and olive oil, as described below. Now, for the validation experiment, the system was increased to a final volume of 30 mL, then increasing the volume to add CS, olive oil, PEITC and TPP.

## 2.3. Optimisation

An experimental strategy was implemented to reduce the number of experiments necessary to optimise the production of CS-olive oil MPs loaded with PEITC, as well as to establish the most influential factors. For this purpose, a 3-level ($3^2$) factorial experimental design was selected, which is a type of 3-level response surface design that includes a subset of runs of a full factorial at 3 levels. The factors evaluated were olive oil: CS mass ratio ($X_A$) and PEITC: CS mass ratio ($X_B$), and the selected responses variable (Y) were the particle size (as Z-average), zeta-potential, polydispersity index, and entrapment efficiency of PEITC. The design resulted in an arrangement of 9 treatments, which was executed in triplicate (a total of 27 runs) on successive days. The levels of the factors, coded as -1 (low), 0 (central point) and +1 (high), are shown in Table 1. To carry out the experiments, we proceeded as established in the exploratory analyses. Immediately after the production of each system, it was analysed, as described below.

## 2.4. MPs characterisation

**2.4.1. Size, zeta-potential and polydispersity index.** The MPs suspensions were analysed concerning their physical properties by dynamic light scattering (DLS). The measured parameters were particle size (Dh), polydispersity index (PdI) and zeta-potential (ZP). Size values for exploratory assays were determined based on intensity distribution by apparent hydrodynamic diameter. In the case of optimisation tests, as the correlation functions were monomodal, the cumulative analysis was applied instead of the distribution method, since it is a simple and robust procedure that does not make any assumption about the size distribution of the sample. The parameters obtained are the mean value for the size (Z-average) and the amplitude of the dispersion curve known as the Polydispersity Index (PdI). The Z-average is a calculated value of diameter based on the % intensity of scattered light by each particle fraction or family and is comparable with other results only if the same scattering angle is used. Therefore, all the Dh values reported in the multifactorial analysis correspond to the Z-average values. The

**Table 1. Levels for 2 experimental factors.**

| Factors | Levels | | |
| :---: | :---: | :---: | :---: |
| | **-1** | **0** | **+1** |
| $X_A$ [1] | 0.00 | 1.25 | 2.50 |
| $X_B$ [2] | 0.25 | 0.75 | 1.25 |

[1] Olive oil: CS mass ratio

[2] PEITC: CS mass ratio.

measurements were made at room temperature (25˚C) using a disposable folded capillary cell (Malvern, Worcestershire, UK), at a constant detection angle of 173˚ using the technology called NIBS (Non-Invasive Back-Scatter) that reduces the effect known as multiple scattering and the influence of contaminants in the sample. The Zetasizer NanoZSP photometric correlation spectrometer (Malvern Instruments Ltd., UK) was used, which allows the measurement of particle sizes between 0.6 nm and 6 μm. This equipment is equipped with a 10 mW He-Ne laser with an emission wavelength of 633 nm. Data were acquired and analysed using Zetasizer v. 7.11 software (Malvern Instruments Ltd., UK). All measures were performed in triplicate.

**2.4.2. Determination of entrapment efficiency.** The PEITC entrapment efficiency (EE) was determined by ultra-filtration centrifugation [21]. 4 mL of MP suspension was placed in a centrifugal filter unit (MWCO 10,000, Amicon® Ultra, Millipore, MA, USA) and centrifuged at 8,000 rpm for 1 h to separate the un-entrapped PEITC. 3 mL of the filtrate containing the free PEITC was concentrated by solid-phase extraction (SPE), using $C_{18}$ Sep-Pak® cartridges (Waters Corporation, Milford, MA USA) with elution in vacuo, using 500 μLpure methanol as elution solvent. Following, the absorbance of the eluates was measured at 245 nm. The same amount of the diluted MP suspension was dissolved in methanol at 60˚C. The resultant sample was filtered through 0.45 μm-membranes and analysed for 245 nm absorbance. To determine the PEITC concentration, a calibration curve was constructed in methanol with PEITC concentrations of 0.2–6.6 mM. Measurements were made in 96-well Nunc UV transparent plate (Thermo Scientific, Waltham, MA, USA) with multidetection plate reader (Synergy H1, Vermont, USA) controlled by the Gen5 Biotek software version 3.04. The amount of PEITC entrapped in the MP was obtained by subtracting the free PEITC amount from the total PEITC amount in the suspension. The EE was calculated as follows:

$$\text{EE}(\%) = \frac{PEITC_{entrapped}}{PEITC_{total}} * 100 \tag{1}$$

**2.4.3. FTIR-ATR analysis.** The spectra of empty and loaded MPs and pure PEITC was obtained with a Fourier transform infrared spectrometer (FTIR) (PerkinElmer Spectrum-100), with a horizontal attenuated total reflectance (ATR) accessory, with a diamond/ZnSe crystal. All spectra were acquired with 30 scans and 4 cm$^{-1}$ resolution, in the region of 4000 to 600 cm$^{-1}$. Three replicates were collected for each sample. Spectrum corresponding to the medium (ultrapure water) was subtracted from the spectra of the MPs suspensions.

## 2.5. *In vitro* biocompatibility

One cell line was considered throughout this work, namely, Caucasian colon adenocarcinoma cells—Caco-2 (ECACC 86010202). Caco-2 cells were maintained in DMEM high glucose supplemented with 10% (v/v) FBS, 1% (v/v) penicillin-streptomycin, and MEM non-essential amino acid solution. The culture was incubated at 37˚C in a 5% (v/v) $CO_2$ humidified atmosphere. Cells were detached using TrypLE Express (Thermo Scientific, Waltham, MA, USA), seeded ($1 \times 10^5$ cells/well) into 96 well Nunc Optical Btm Plt PolymerBase Black microplates (Thermo Scientific, Waltham, MA, USA), and incubated for 24 h. Afterwards, the culture media were carefully removed and replaced with empty MPs, PEITC loaded MPs and pure PEITC at different equivalent PEITC concentrations (38, 77, 153 and 306 μM for MPs; 51, 102, 204 and 408 μM for pure PEITC), all sterile filtered. It is worth clarifying the differences in the concentrations, because we carry out serial dilutions, on the one hand from pure PEITC and on the other from the encapsulated concentration. After incubation for 24 h, the cytotoxicity of the samples was evaluated using the PrestoBlue™ HS Cell Viability assay (Thermo Scientific,

Waltham, MA, USA), following the protocol described by the manufacturer. Fluorescence was measured using a fluorescence excitation wavelength of 560 nm and an emission of 590 nm by a microplate reader (Synergy H1, Biotek Instruments, Winooski, VT, USA). Cells in culture medium were used as a control, and wells without cells were used as blanks. The metabolic inhibition was determined according to the following Eq (2):

$$\% \text{ Metabolic Inhibition} = \frac{F_{control} - F_{sample}}{F_{control}} * 100 \tag{2}$$

where $F_{control}$ and $F_{sample}$ are the fluorescence intensities at 590 nm of control and sample, respectively. Six replicates for each condition were performed (n = 5).

## 2.6. Statistics

Each optimisation experiment was performed in triplicate and the results were expressed as the mean values with standard deviations (SD). For each response (Y), i.e. the previously explained responses from each treatment, the normal distribution of the residuals was determined by graphical analysis. Likewise, using the Durbin-Watson statistic, we verified for each model that the residuals did not present any significant correlation based on the order of the data. Then, the data were fitted to the following polynomial quadratic model:

$$Y = \beta_0 + \beta_A X_A + \beta_B X_B + \beta_{A,A} X_A^2 + \beta_{A,B} X_A X_B + \beta_{B,B} X_B^2 + \varepsilon \tag{3}$$

where $X_A$ and $X_B$ are the coded levels of the independent variables mentioned above; $\beta_0$, $\beta_i$, $\beta_{i,i}$ and $\beta_{i,j}$ are the regression coefficients for the independent term, the linear, quadratic and binary interaction effects respectively; and $\varepsilon$, the residual error [22,23]. The surface and contour graphs of the responses were generated from this model. The significance of the effects of each experimental factor in the model was estimated for each analysed response. The models were recalculated, considering only the significant effects, and thus obtained regression models with a good explanation of the data variability ($R^2$). Finally, a multi-criterion optimisation based on Derringer's desirability function was applied [24] to the results of the experimental design, expressing the desirability of each response value on a scale of 0–1.

In the case of the validation test, a two-tailed hypothesis test relative to the difference between two sample means from normal distributions was applied for each response, considering a significance of 0.05 to reject the null hypothesis.

All statistical analysis was carried out with the aid of RStudio V 1.2.1335.

## 3. Results

### 3.1. Exploratory analysis

**3.1.1. Effect of ultrasound homogenisation.**  In this work, CS MPs loaded with PEITC were formulated. Before expanding the formulation of these systems and their optimisation, we needed to set some technical parameters. As we intended to use ultrasound homogenisation (UH), it was necessary to determine when to homogenise and then how much. Thus, we tried to produce the systems without ultrasound homogenisation (without UH); with homogenisation after ionic gelation (simple UH); and before and after ionic gelation (dual UH). In all cases, magnetic stirring was applied at high revolutions during the ionic gelation dripping. Fig 2 shows the distributions of the apparent hydrodynamic sizes (Dh) of the particles obtained in each condition. So, we saw that applying dual UH gave better results. Then, to continue, we evaluated the homogenisation time, maintaining the intensity at 70% (parameter optimised in previous works). We tried homogenising for 30 s, 45 s, 60 s and 120 s. We saw that it was more

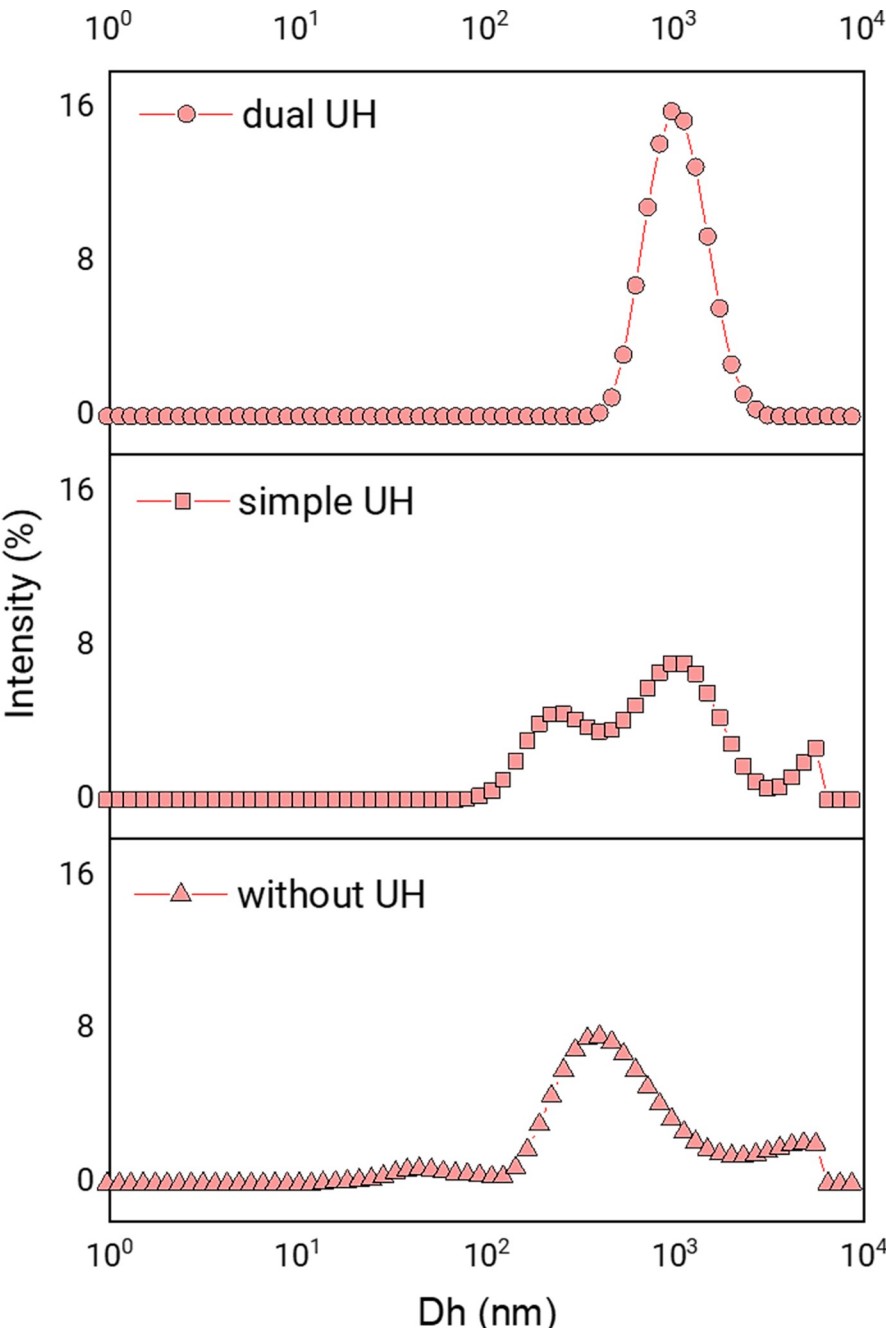

**Fig 2. Effect of ultrasound.** Distribution of apparent hydrodynamic diameters (Dh) according to the percentage of dispersion intensity, showing the effect of homogenisation with ultrasound in the formation of CS MPs. Results corresponding to the process with homogenised pre- and post-ionic gelation (dual UH); homogenised before ionic gelation (simple UH); without homogenised with ultrasound (without UH).

reproducible from 60 s. At 120 s, the system already acquired a very high temperature if it was not cooled. Therefore, we decided to establish the time of 60 s as a constant parameter of the process.

**3.1.2. Effect of TPP flow rate.** Another parameter to be analysed in the exploratory study was the drip flow of TPP in ionic gelation. For this, flows of 50, 100, 200 and 250 μL/min were

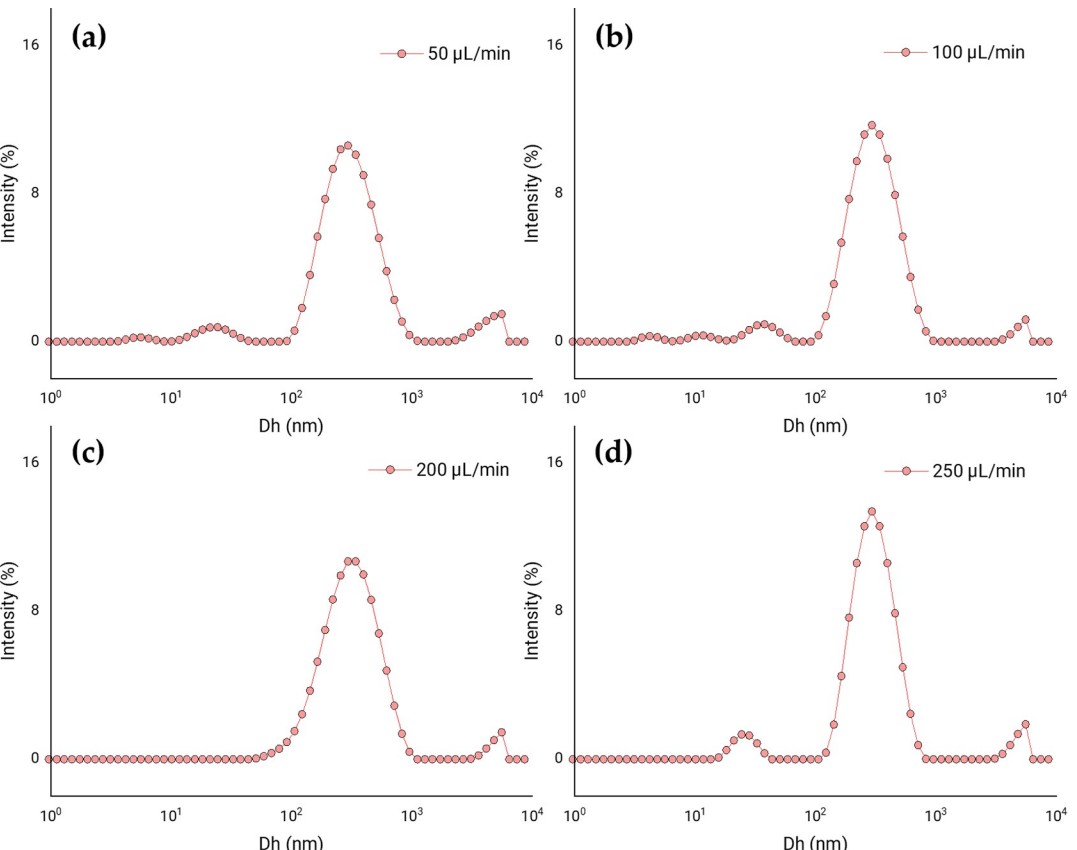

**Fig 3. Effect of TPP dripping.** Distribution of apparent hydrodynamic diameters (Dh) according to the percentage of dispersion intensity, showing the effect of the TPP flow rate on the ionic gelation of CS MPs with PEITC. Results corresponding to the process with homogenisation with ultrasound pre- and post-ionic gelation, and with a drop rate of TPP of 50 μL/min (**a**); 100 μL/min (**b**); 200 μL/min (**c**); 250 μL/min (**d**).

tested to polymerise MPs loaded with PEITC. Fig 3 shows the size distribution of the particles obtained in each condition. We saw that there were no appreciable changes in the distributions with increasing flow. However, due to the needle used for droplet formation, at a flow rate of 250 μL/min, some drops were projected out of the polymerisation vessel due to the increased pressure of the syringe plunger. Therefore, a flow rate of 200 μL/min was chosen as a constant for all subsequent experiments.

**3.1.3. Incorporation of olive oil.** CS MPs were produced with olive oil in a ratio of 1.25:1.00 (olive oil: CS mass ratio): without PEITC and loaded with PEITC (0.50:1.00 PEITC: CS mass ratio). Fig 4 shows the particle size distributions in both conditions. As the graphs show, the incorporation of a liquid lipid such as olive oil improved the formation of polymeric MPs loaded with PEITC. Furthermore, the system was better when both olive oil and PEITC were present (Fig 4a) than when PEITC was not loaded (Fig 4b).

## 3.2. Optimisation

Once the exploratory analyses have been completed, and the interest in producing hybrid MPs (polymeric-lipid) with CS and olive oil to load PEITC has been established, we continued to develop an optimal model for our new system. Considering this, we selected an experimental design of the 3-level ($3^2$) factorial type (described in section 2.3; see S1 Appendix for raw data and full analysis). We selected the factor levels considering that the important thing was the

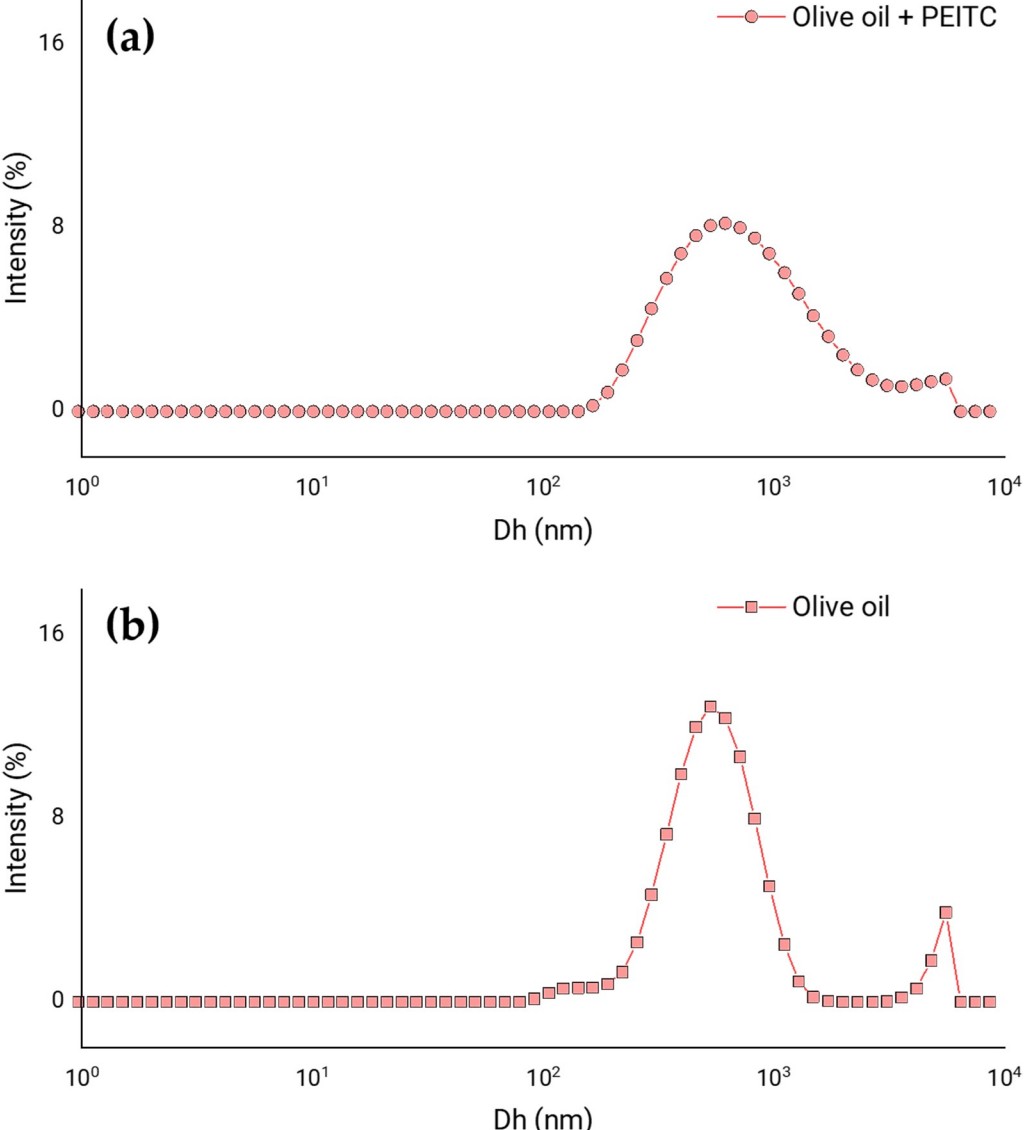

**Fig 4. Effect of olive oil.** Distribution of apparent hydrodynamic diameters (Dh) according to the percentage of dispersion intensity, showing the effect of the incorporation of olive oil in the formulation of CS MPs loaded with PEITC. Results corresponding to the MPs with a mass ratio of olive oil: CS of 1.25 to 1.00, with PEITC (**a**) and without PEITC (**b**).

presence of PEITC, while the absence of olive oil would be considered. Table 2 shows the design matrix and the multiple regression results obtained for the 9 randomised treatments.

Fig 5 shows the Pareto charts with the contribution of the standardised effects of the model coefficients, for each response considered. Broadly speaking, we can say that in the case of the particle size (Fig 5A) and the polydispersity index (Fig 5C), the quadratic effect of the PEITC concentration was not significant, which could be neglected. In the case of the zeta-potential (Fig 5B), two coefficients did not have a significant effect on the response, both being quadratic coefficients. This means that the surface modelled for that response would have no curvature. For the entrapment efficiency (Fig 5D), the quadratic coefficient of the olive oil concentration did not have significance, thus it was negligible for the model.

**Table 2. Central composite factorial design for two factors and four responses.**

| Run | Factors | | Responses [†] | | | |
|---|---|---|---|---|---|---|
| | $X_A$ | $X_B$ | Dh [1] | ZP [2] | PdI [3] | EE [4] |
| 1 | 2.50 | 0.75 | 729 ± 19 | 10.6 ± 1.9 | 0.287 ± 0.014 | 99.46 ± 0.11 |
| 2 | 1.25 | 0.75 | 663 ± 3 | 22.1 ± 1.5 | 0.267 ± 0.009 | 99.55 ± 0.08 |
| 3 | 0.00 | 0.25 | 744 ± 87 | 33.7 ± 3.1 | 0.618 ± 0.042 | 98.21 ± 0.05 |
| 4 | 1.25 | 1.25 | 677 ± 18 | 13.2 ± 2.2 | 0.265 ± 0.002 | 99.65 ± 0.12 |
| 5 | 0.00 | 1.25 | 2222 ± 63 | 31.6 ± 1.5 | 0.299 ± 0.017 | 99.63 ± 0.08 |
| 6 | 2.50 | 1.25 | 735 ± 17 | 5.7 ± 0.1 | 0.274 ± 0.012 | 99.65 ± 0.04 |
| 7 | 0.00 | 0.75 | 1436 ± 194 | 37.5 ± 0.4 | 0.487 ± 0.016 | 99.44 ± 0.03 |
| 8 | 2.50 | 0.25 | 674 ± 40 | 19.8 ± 3.7 | 0.278 ± 0.008 | 99.08 ± 0.08 |
| 9 | 1.25 | 0.25 | 624 ± 5 | 32.3 ± 1.0 | 0.275 ± 0.010 | 98.70 ± 0.31 |

[1] Z-average diameter in nm

[2] zeta-potential in mV

[3] polydispersity index

[4] entrapment efficiency in %

[†] Values expressed as mean ± SD of three replicates.

Then, we recalculated the models, by eliminating the non-significant effects, with the criterion of maximizing the $R^2_{adj}$, fitting again from the multiple regression of the data. Table 3 shows the parameters of the final models, including the obtained regression statistics: $R^2$, $R^2_{adj}$ and RSD. The new models adequately explained the variation in the response, with a maximum $R^2_{adj}$, all explaining more than 86% of the variability. Fig 6 shows the response surfaces of the final models, expressing the four responses as a function of the two considered factors.

Fig 6A represents the response surface for the Z-average diameter. We saw that the size of the particles produced grew towards maximum values of the PEITC content and minimum values of olive oil. This increase accelerated abruptly close to these limit values. At the same time, this size decreased with the decrease in the PEITC concentration, reaching its minimum at intermediate values of the range tested for the olive oil content. For its part, the zeta-potential (Fig 6B) increased, as expected, by decreasing the content of both hydrophobic compounds (PEITC and olive oil). As we already mentioned in the analysis of Pareto charts, this response surface was flat, so that, just as it reached its maximum in one of its vertices, it reached its minimum in another, at the maximum concentration values of PEITC and olive oil. Fig 6C shows the PdI reached maximum values at minimum values of PEITC and olive oil. While at high concentrations of olive oil the PEITC concentration hardly affected the PdI, which was low, it reached its minimum at high concentrations of PEITC and average values of olive oil. Finally, Fig 6D shows the response surface for entrapment efficiency. We saw that the EE was minimal at low PEITC and olive oil values. This surface presents a marked curvature for the PEITC concentration, so it reached almost maximum EE values already at intermediate values of the PEITC concentration range, beyond the olive oil concentration. However, the EE maximum was reached at the apex of the PEITC maximum and olive oil minimum. The desired optimisation for particle size and polydispersity index was minimisation, while for the case of zeta-potential and entrapment efficiency it was maximisation.

The optimal values for each response as well as the values of the factors for each of these optimal ones can be seen in Table 4. After the responses were optimised separately, they were optimised together based on the 'Derringer desirability' function. For this purpose, the goals of each of the responses were currently established as:

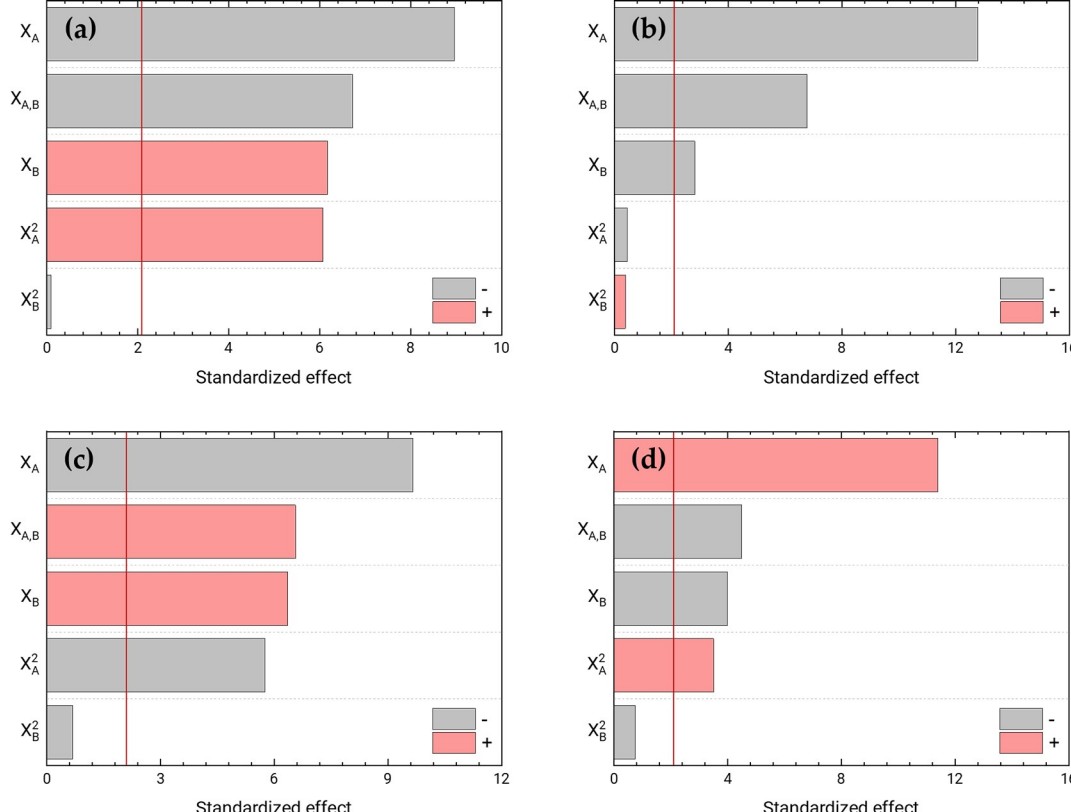

**Fig 5. Pareto charts.** Analysis of the effects of the central composite factorial design. Pareto charts with standardised effects of two experimental factors, in decreasing order of importance (in absolute value) for the Z-average diameter (**a**), zeta-potential (**b**), PdI (**c**), and entrapment efficiency (**d**). The vertical red lines represent the threshold of significance ($P = 0.05$) for 20 degrees of freedom.

**Table 3. Estimates of the coefficients of each term in the recalculated model and the corresponding statistics.**

| Coefficient | Estimated coefficient values | | | |
| --- | --- | --- | --- | --- |
| | Dh [1] | ZP [2] | PdI [3] | EE [4] |
| $\beta_0$ | 566.958 | 38.4027 | 0.669042 | 97.5416 |
| $\beta_A$ | -594.056 | -5.28433 | -0,337033 | 0.376989 |
| $\beta_B$ | 1214.28 | -578389 | -0.2685 | 3.36528 |
| $\beta_{A,A}$ | 281.778 | - | 0.0669511 | - |
| $\beta_{B,B}$ | - | - | - | -1.30556 |
| $\beta_{A,B}$ | -555.067 | -4.812 | 0.126 | -0.342 |
| **Statistics** | | | | |
| $R^2$ | 0.908 | 0.910 | 0.913 | 0.903 |
| $R^2_{adj}$ | 0.891 | 0.899 | 0.898 | 0.885 |
| RSD | 173 | 3.559 | 0.040 | 0.170 |

[1] Z-average diameter

[2] zeta-potential

[3] polydispersity index

[4] entrapment efficiency.

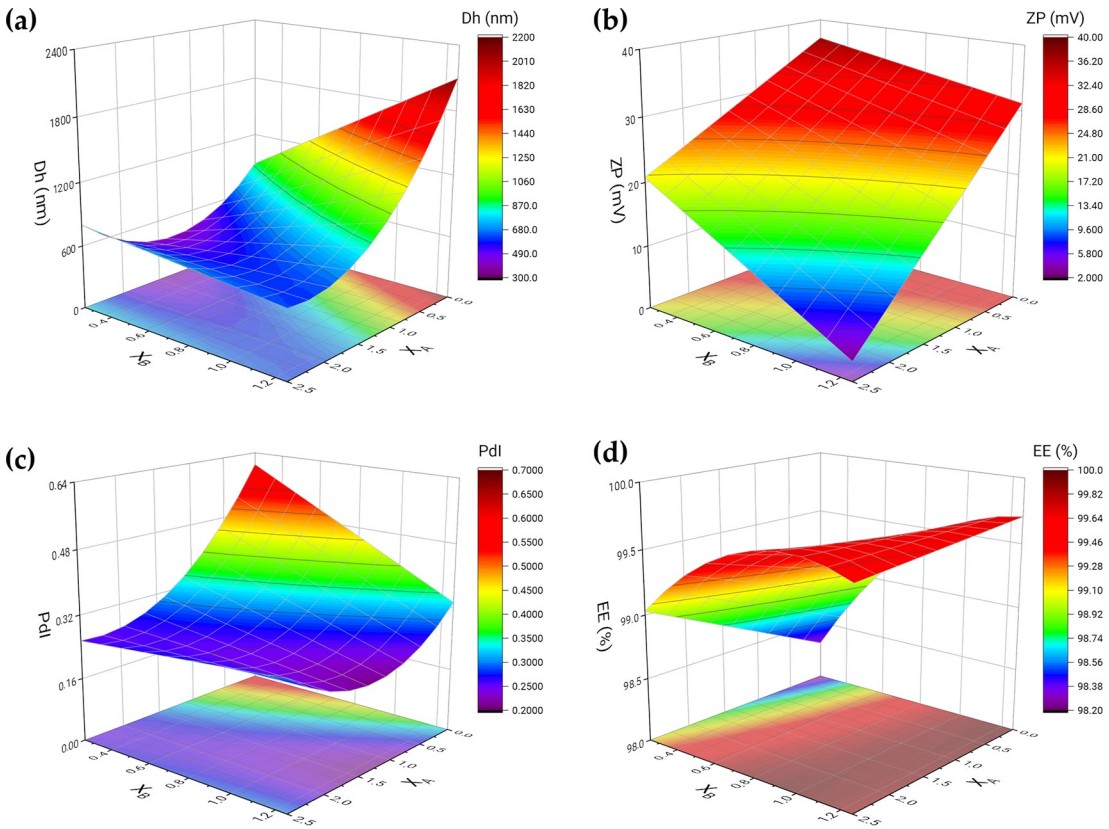

**Fig 6. Response surfaces.** Response surface models for the responses estimated based on two experimental. Charts for Z-average diameter (**a**), zeta-potential (**b**), PdI (**c**), and entrapment efficiency (**d**).

- Dh–minimise between 300–700 nm

- ZP—different from 0 mV

- PdI–minimise

EE was not considered for global optimisation, since it was not a determining factor since extremely satisfactory values were obtained in all experiments. Table 5 shows the combination

**Table 4. Values of the factors that optimise each response and the corresponding predicted optimal response value.**

| Model | Factor | | Response [3] |
|---|---|---|---|
| | $X_A$ [1] | $X_B$ [2] | |
| Dh | 1.30 | 0.25 | 394 |
| ZP | 0.00 | 0.25 | 37.0 |
| PdI | 1.34 | 1.25 | 0.213 |
| EE | 0.00 | 1.25 | 99.71 |

[1] Olive oil: CS mass ratio

[2] PEITC: CS mass ratio

[3] Responses: dh (Z-average diameter) in nm; ZP (zeta-potential) in mV; PdI (polydispersity index) dimensionless; EE (entrapment efficiency) in %.

**Table 5. Derringer optimisation of polymeric MPs and validation.**

| Factor | Low | High | Optimal |
|---|---|---|---|
| $X_A$ [1] | 0.00 | 2.50 | 1.45 |
| $X_B$ [2] | 0.25 | 1.25 | 0.25 |
| **Optimal desirability:** 0.849 | | | |
| Response [3] | Optimal predicted [†] | | Observed [†] |
| Dh (nm) | 400 ± 323 | | 629 ± 50 |
| ZP (mV) | 27.6 ± 6.5 | | 32.3 ± 1.75 |
| PdI | 0.300 ± 0.073 | | 0.329 ± 0.040 |
| EE (%) | 98.72 ± 0.31 | | 98.49 ± 0.09 |

[1] Olive oil: CS mass ratio

[2] PEITC: CS mass ratio

[3] Responses: Dh (Z-average diameter) in nm; ZP (zeta-potential) in mV; PdI (polydispersity index) dimensionless;
EE (entrapment efficiency) in %.

[†] Values expressed as mean ± SD of three replicates.

of factor levels that maximises the function of 'desirability' in the indicated region. This table also shows the predicted values for each response in that global optimal condition.

Likewise, Table 5 shows the result of the validation of the model as "observed". For this validation, the predicted optimal formulation was produced but with additional complexity. The variability of increasing the volume of the reaction system by three times was added to the experiment, maintaining the same ultrasound probe as well as the same syringe and needle. Comparing the value of each response between predicted and observed with the hypothesis test, with a significance level of 0.05, we found no statistical differences, for which we were able to validate our model. Subsequent tests were carried out with this formulation.

### 3.3. FTIR

Fig 7 shows the infrared spectra of the MPs and pure PEITC. The characteristic peaks of the isosulphocyanic group (N = C = S) of PEITC [25] changed when incorporated into the MP. The respective bands of asymmetric and symmetric N = C = S stretching, changed from 2182 cm$^{-1}$ and 2083 cm$^{-1}$ to 2188 cm$^{-1}$ and 2092 cm$^{-1}$, respectively in MP + PEITC. The low wave-number shift revealed that the strength of the double bonds (N = C = S) was altered due to the occurrence of the inclusion action. A series of PEITC absorbance peaks disappeared or weakened with encapsulation as follows: -H stretch vibration at 3028 cm$^{-1}$ in the benzyl group; bending vibration of -CH$_3$ at 1454 and 1347 cm$^{-1}$; C-H plane bending vibration in benzyl around 700 cm$^{-1}$. A typical characteristic of the crosslinking between CS and TPP can also be seen in both MPs. Wu et al. [26] established that the peak at 1610 cm$^{-1}$ of the bending vibration of–NH$_2$, characteristic of CS, shifts to 1532 cm$^{-1}$ in the polymerisation between CS and TPP. In more recent work, Costa et al. [27] saw that same band at 1586 cm$^{-1}$. In our case, we saw this peak at 1555 cm$^{-1}$ for MP, a shift perhaps due to the presence of olive oil. Furthermore, in the case of MP + PEITC, there was a smaller shift, at 1552 cm$^{-1}$, perhaps due to the inclusion of PEITC. We also found two bands related to the acetate in the medium [28,29]. The 1745 cm$^{-1}$ (absorption peak of the carboxyl group) and 1416 cm$^{-1}$ ($\delta$C-H vibrations of the CH$_3$C = O residue) bands in the MP shifted to 1744 cm$^{-1}$ and 1417 cm$^{-1}$ in the MP + PEITC, respectively. In addition to displacement, there was a decrease in the intensity of the 1417 cm$^{-1}$ band and an increase in the 1744 cm$^{-1}$ band in MP + PEITC compared to MP. The increase in intensity to

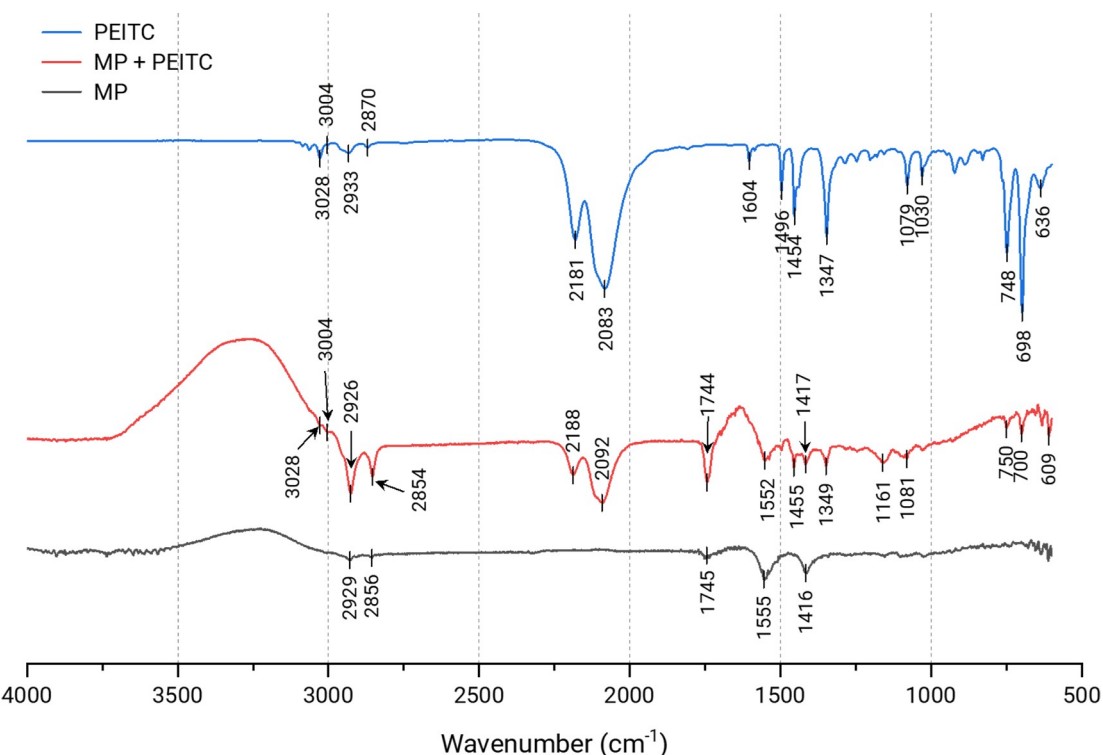

**Fig 7. Fourier transform infrared transmittance spectra for empty MPs, MPs with PEITC (MP + PEITC) and pure PEITC.**
FTIR.

1744 cm$^{-1}$ implies an increase in the number of free acetate ions, while the intensity of the band related to the carboxyl salt (1417 cm$^{-1}$) decreases [30]. The results indicate that the electrostatic interactions between the carboxyl group of the acetate and the amino groups of the CS were altered with the inclusion of PEITC.

## 3.4. Biocompatibility

The biocompatibility (or cytotoxicity) of the optimal MPs with and without PEITC, as well as for pure PEITC, was analysed. For this, metabolic inhibition against Caco-2 cells was measured. According to the international standard ISO 10993–5 for the biological evaluation of medical devices (part 5: Tests for *in vitro* cytotoxicity), the threshold value for a sample to be cytotoxic is a metabolic inhibition of 30%. The results of the biocompatibility test were expressed as a function of the equivalent concentration of PEITC, given that although the empty MPs did not present PEITC, they were diluted in the same proportion as those loaded with PEITC. As Fig 8 shows, pure PEITC at 408 μM was cytotoxic, while below 204 μM it had no cytotoxicity. In fact, below 102 μM the PEITC showed negative inhibition values, which could be due to the reported ability to stimulate antioxidant processes in the cell itself, increasing the reducing environment [31], which is precisely what the PrestoBlue assay directly measures. In the case of the MPs without PEITC, they showed constant negative inhibition values despite the concentration, which may also be related to the reducing power of CS [32]. For the PEITC-loaded MPs, cytotoxicity was already observed at the 153 μM PEITC concentration, while the inhibition values were negative below 77 μM. Thus, it turned out that encapsulated PEITC presented greater inhibitory power against Caco-2 cells than free PEITC.

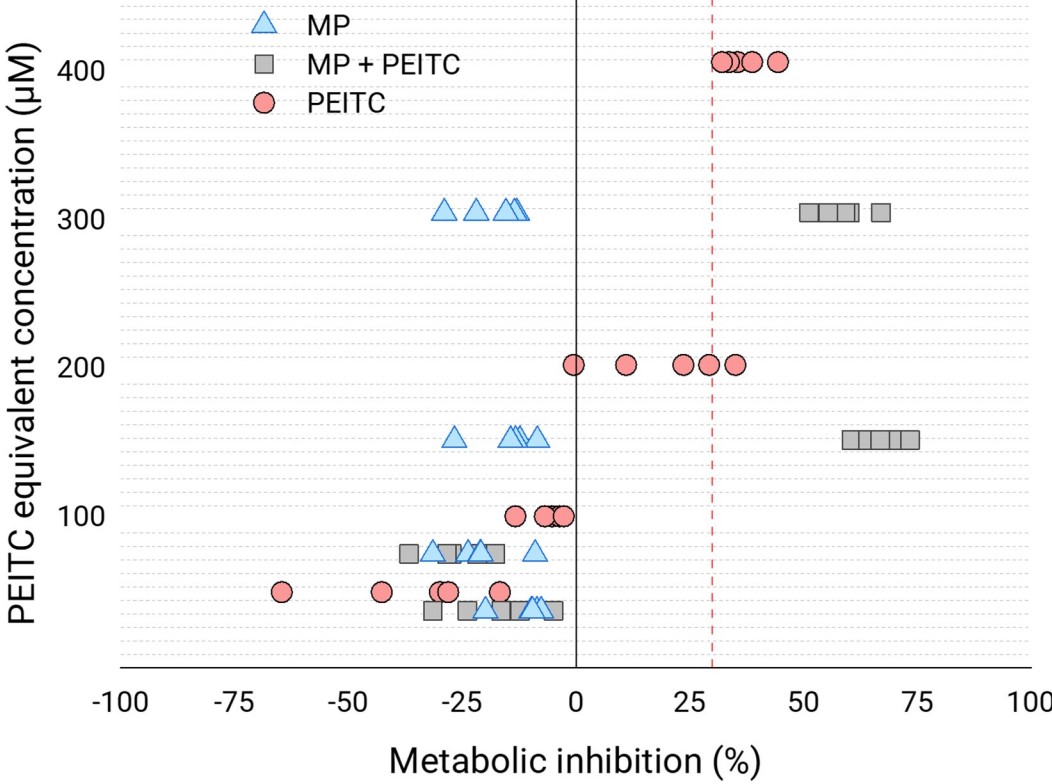

**Fig 8. Biocompatibility.** Metabolic inhibition of empty MPs, MPs with PEITC (MP + PEITC) and pure PEITC at different equivalent concentrations of PEITC (38, 77, 153 and 306 μM for MP and MP + PEITC; 51, 102, 204 and 408 μM for pure PEITC) for 24 h against human colorectal adenocarcinoma Caco-2 cells. The scatter of independent determinations (n = 5) are represented. The dashed red vertical line represents the 30% inhibition limit.

## 4. Discussion

According to the National Nanotechnology Initiative, nanoparticles range from 1 to 100 nm. Meanwhile, the MPs can vary between 100 and $10^5$ nm. However, in pharmaceutical applications, the most common size varies between 100 and 1000 nm [33]. In order not to diffuse into the bloodstream or interstitial space and thus guarantee its arrival at the target site, the size of the MPs should not be less than 300 nm. In this sense, in this work, it was established that the ideal size of the MPs would be between 300 and 1000 nm. Therefore, it is essential to evaluate this parameter, to ensure that the MPs comply with the desired sizes since their dimensions interfere with the biodistribution, toxicity and bioavailability of the organism [17]. In the case of this study, this restriction was met by almost all the systems produced throughout the trials, highlighting this property for the case of the optimal system developed. With this, we can say that we obtained particles of satisfactory size.

Zeta-potential is an indicator that predicts the stability of MPs in suspension, their tendency to aggregate, noting that particle aggregation compromises bioaccessibility to target sites [18]. This parameter expresses the potential difference at the interface of the surface of the MP and the solvent, so a zeta-potential between 20–30 mV guarantees moderate stability of the particles [15,20]. CS is a mucoadhesive polymer since its surface is full of positive charges. Therefore, the zeta-potential of MPs based on this polymer will be positive, indicating that, initially, the CS particles will be stable in suspension. Thus, the zeta-potential can predict the *in vivo* behaviour of these particles when they come into contact with biological media [13].

Furthermore, the zeta-potential can help to assess whether the PEITC is well encapsulated within the particle or whether it was only adsorbed on the surface [34]. In our case, we observe that the zeta-potential decreases with the increase in PEITC and olive oil (uncharged, hydrophobic), with which part of it is adsorbed on the surface, but continues at more than acceptable values and, what is more, in the optimal system the value exceeds 30 mV. From this, it can be thought that the reticulated interior manages to create an even more hydrophobic and fluid environment with the presence of the liquid lipid. With this, we can say that we obtained particles not only with a satisfactory size but also stable and that they manage to incorporate both the lipid that acts as a copolymer and the compound to be loaded.

A priori, the ideal proportion of PEITC to add to the MPs had to be tested based on the effectiveness of encapsulation, so it is known that as more PEITC is incorporated, after a certain point, and EE begins to decrease, possibly due to saturation of the MPs [35]. However, in our case, we observed a very high EE in all cases. These high values probably did not correspond only to the trapped PEITC, but also to the adsorbed one, as we can see from the decrease in zeta-potential, given the great hydrophobicity of PEITC. Furthermore, we can reinforce this idea with FTIR analyse. The FTIR results indicate that the inclusion process affected the optical behaviour of the PEITC, as well as the MP itself and its interaction with the medium, which may point in the same direction as explained in the previous lines.

Knowing the dosages of a product is essential for its formulation since it is essential to know what quantities should be ingested for a specific purpose. On the other hand, these doses must first be biocompatible, that is, they must not present cytotoxicity. In a 2018 study, humans were given a dose of 40 to 80 mg PEITC/day orally for 30 days, with no adverse effects. However, when the consumed dose was increased to values between 120 and 160 mg of PEITC/day, during the same experimental period, some toxicity was observed [36]. In other studies, it was concluded that the acceptable daily dose is only 40 mg in humans, which is consistent with the previous study [5]. According to Abbaoui et al. [37], despite having some toxic effects on organisms, the dose for PEITC to exert its preventive and therapeutic effect is nontoxic and safe for humans. By analysing different reports, a range of PEITC performance values can be established from the most preventive phase to the most therapeutic phase, ranging from 5–30 μM, corresponding to 0.82–4.90 mg/L [1,38–43]. From our biocompatibility results, we can verify that the doses with therapeutic activity would be safe, both for free and encapsulated PEITC. Now, just as we saw that the toxicity of encapsulated PEITC was presented at lower concentrations than free, it remains to assess whether the same thing happens with preventive/therapeutic effectiveness. To do this, other parameters such as the PEITC release rate must first be analysed, as discussed in the next paragraph.

This work results in a novel advance about ITCs as therapeutics. This is due to the novelty of encapsulating them, advancing technologically in preparation for possible biomedical and nutraceutical applications. This is a journey that is just beginning. What comes next? We still must analyse if our encapsulated compound is properly released, and if it is released how and how much. We must also analyse how encapsulation influences its bioactivities. Next, we must study the stability under gastrointestinal conditions. And finally, within our scope, to study whether CS improves the adhesion of these MPs and release in the lesion regions of the colon, where there is a greater expression of mucins.

## Supporting information

**S1 Appendix. Raw data and full statistical analysis of the 3-level ($3^2$) factorial experimental design.**
(DOCX)

## Author Contributions

**Conceptualization:** Ezequiel R. Coscueta, Celso A. Reis, Manuela Pintado.

**Data curation:** Ezequiel R. Coscueta.

**Formal analysis:** Ezequiel R. Coscueta.

**Funding acquisition:** Manuela Pintado.

**Investigation:** Ezequiel R. Coscueta.

**Methodology:** Ezequiel R. Coscueta.

**Project administration:** Celso A. Reis, Manuela Pintado.

**Resources:** Ezequiel R. Coscueta, Celso A. Reis, Manuela Pintado.

**Software:** Ezequiel R. Coscueta.

**Supervision:** Celso A. Reis, Manuela Pintado.

**Validation:** Ezequiel R. Coscueta.

**Writing – original draft:** Ezequiel R. Coscueta, Ana Sofia Sousa.

**Writing – review & editing:** Ezequiel R. Coscueta, Celso A. Reis, Manuela Pintado.

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
