## [Decision Letter · Decision Letter 0]

22 Mar 2021

PONE-D-21-05688

Chitosan-olive oil microparticles for phenylethyl isothiocyanate delivery: optimal formulation

PLOS ONE

Dear Dr. Coscueta,

Thank you for submitting your manuscript to PLOS ONE. After careful consideration, we feel that it has merit but does not fully meet PLOS ONE’s publication criteria as it currently stands. Therefore, we invite you to submit a revised version of the manuscript that addresses the points raised during the review process.

We look forward to receiving your revised manuscript.

Kind regards,

Vivek Gupta

Academic Editor

PLOS ONE

Journal Requirements:

Reviewers' comments:

Reviewer's Responses to Questions

**Comments to the Author**

1. Is the manuscript technically sound, and do the data support the conclusions?

Reviewer #1: No

Reviewer #2: Yes

Reviewer #3: Yes

2. Has the statistical analysis been performed appropriately and rigorously? 

Reviewer #1: No

Reviewer #2: Yes

Reviewer #3: Yes

3. Have the authors made all data underlying the findings in their manuscript fully available?

Reviewer #1: No

Reviewer #2: Yes

Reviewer #3: Yes

4. Is the manuscript presented in an intelligible fashion and written in standard English?

Reviewer #1: No

Reviewer #2: Yes

Reviewer #3: Yes

5. Review Comments to the Author

Reviewer #1: In the study, author attempted to optimized microparticles of phenylethyl isocyanate. There is major design flaws in the paper and it cannot be accepted in the current state. Please see below for current format.

Comment 1: The writing required to be improved throughout the manuscript. The writing is non cohesive and unclear at times. Writing required to be improved significantly.

Comment 2: Factorial design was used for optimization of formulation. In the study, olive oil to chitosan and PEITC to chitosan ration were two independent variables. However, these variables are not independent of each other i.e. varying the ratio of chitosan would affect both variables. Also, page # 14 refer the design as central composite design. It is not clear which design was used in the study.

Comment 3: It is not clear why olive oil was incorporated in formulation.

Comment 4: To determine the significant variable, ANOVA tablet must be included. It is not possible for the reader to conclude if the factor is significant without ANOVA table. Line 453-547 mention the factors are significant; however, it is not possible to confirm without ANOVA table.

Comment 5: All particle size graphs don’t have 100% intensity; Y-axis should be explained in the manuscript. In addition, dilution factor used for PSD is not mentioned in the paper.

Comment 6: Preparation method for microparticles is not incorporated in the manuscript.

Comment 7: The formulation composition is not shown in the table.

Reviewer #2: 1. Page 10, Line 211: Please check and correct the cell number for right representation.

2. Page 10, Line 214-216: The tested concentrations for different treatments were different. Please explain and make it clear in these lines.

3. Please introduce design of experiments in the introduction section.

4. Page 20, Section 4: In-vitro biocompatibility studies: Two kinds of fluorescence measurements have been described in the methods section. Please clearly distinguish and elaborate on the fluorescence measurements at excitation wavelength of 560 nm and an emission of 590 nm; the fluorescence intensities at 485 nm. Only % metabolic inhibition was represented in the results and figures.

5. Authors should clearly state the optimal formulation chosen for all the characterization studies carried out.

6. Please include the importance of central composite design in the discussion by citing any previous works.

7. It is also recommended to cite any reports referring to the safe use of microparticle components especially highlighting the dosage.

8. English needs to be thoroughly checked and can be improved at many places in the manuscript.

Reviewer #3: COMMENTS TO AUTHOR

1. Authors have to give the clarity regarding type of route of administration.

2.Authors have to provide stability data of microparticles as a supplement data.

3.what is the charge of PEITC? Justify its effect on stability and entrapment.

4.Author have to justify use of lower level of XA variable. Why 0 value as a lower level. If polymer concentration is zero than how microparticles will form?

Justify use of olive oil in formulations.

Overall work is good and authors have to correc spacing formating as per author instructions.

6. PLOS authors have the option to publish the peer review history of their article (what does this mean?). If published, this will include your full peer review and any attached files.

Reviewer #1: No

Reviewer #2: **Yes: **Vineela Parvathaneni

Reviewer #3: **Yes: **Dr.Deepa Patel

---

## [Author Response · Author response to Decision Letter 0]

22 Mar 2021

Dear Reviwers, 

Many thanks to you for all the inputs and valuable suggestions that will contribute to improve our manuscript significantly. 

The answers are given just after the transcription of reviewers’ comments. New information and corrections added to the new manuscript and they are marked in the “Revised Manuscript with Track Changes” file.

Reviewer #1: In the study, author attempted to optimized microparticles of phenylethyl isocyanate. There is major design flaws in the paper and it cannot be accepted in the current state. Please see below for current format.

Comment 1: The writing required to be improved throughout the manuscript. The writing is non cohesive and unclear at times. Writing required to be improved significantly.

Response: Thank you very much for your comment, which has been considered and the full text has been revised.

Comment 2: Factorial design was used for optimization of formulation. In the study, olive oil to chitosan and PEITC to chitosan ration were two independent variables. However, these variables are not independent of each other i.e. varying the ratio of chitosan would affect both variables. Also, page # 14 refer the design as central composite design. It is not clear which design was used in the study.

Response: The reviewer is correct, and this was always considered as can be seen in the design itself when we specified in line 139 “(…) the process began with the addition of 4 mL of CS (…)”, which means that the concentration of chitosan was a constant in all experiments. As perhaps it was not entirely clear, now we highlight it in the text.

Comment 3: It is not clear why olive oil was incorporated in formulation.

Response: Lines 64-66 say "Besides, PEITC was already stabilised with vegetable oils, such as olive oil, once vegetable oil protects non-polar ITCs from decomposition or volatilisation." This may have gone very unnoticed in the text, which is why it is now reinforced between lines 94-96 of the introduction, in which the objectives of the work are highlighted.

Comment 4: To determine the significant variable, ANOVA tablet must be included. It is not possible for the reader to conclude if the factor is significant without ANOVA table. Line 453-547 mention the factors are significant; however, it is not possible to confirm without ANOVA table.

Response: As mentioned in line 323 all the data and statistics of the experimental design are available in the S1 Appendix.

Comment 5: All particle size graphs don’t have 100% intensity; Y-axis should be explained in the manuscript. In addition, dilution factor used for PSD is not mentioned in the paper.

Response: The intensity distribution is naturally weighted according to the scattering intensity of each particle fraction or family, therefore 100% is obtained by integrating the area under the curve. According to the reviewer's request, we add a few exclamatory words in the methodology. For more information you can consult the link: 

https://www.materials-talks.com/blog/2017/01/23/intensity-volume-number-which-size-is-correct/

To carry out the measurements of size and zeta-potential there was no dilution, it was done directly on the formed system.

Comment 6: Preparation method for microparticles is not incorporated in the manuscript.

Response: We regret not agreeing with the reviewer, but that is not correct, Section “2.2. Microparticle production” specifies the entire preparation of the microparticles.

Comment 7: The formulation composition is not shown in the table.

Response: The formulation is what is optimised, that is why it is well reported, and even in Table 5 the optimal composition is shown.

Reviewer #2: 

1. Page 10, Line 211: Please check and correct the cell number for right representation.

Response: Thank you very much for noticing the formatting error, it is fixed in the new version.

2. Page 10, Line 214-216: The tested concentrations for different treatments were different. Please explain and make it clear in these lines.

Response: We appreciate the comment, it is clarified in the new version.

3. Please introduce design of experiments in the introduction section.

Response: Thank you very much for the comment, in the new version we mention the experimental design on line 112 of the introduction.

4. Page 20, Section 4: In-vitro biocompatibility studies: Two kinds of fluorescence measurements have been described in the methods section. Please clearly distinguish and elaborate on the fluorescence measurements at excitation wavelength of 560 nm and an emission of 590 nm; the fluorescence intensities at 485 nm. Only % metabolic inhibition was represented in the results and figures.

Response: Thank you very much for noticing the error. It was really a typing error, because we use the same formula for XTT protocol (which is cheaper, but PrestoBlue is more sensitive) which absorbs at 485 nm.

5. Authors should clearly state the optimal formulation chosen for all the characterization studies carried out.

Response: The optimal formulation was specified in Table 5. In line with the reviewer's request, a clarification was added on line 411.

6. Please include the importance of central composite design in the discussion by citing any previous works.

Response: We appreciate the comment, unfortunately in this case we do not agree. The factorial design, at least in this work, is hardly a statistical tool, commonly used in all engineering for formulations. That is why we think that mentioning its importance in the discussion is like talking about any of the other analytical methodologies used, that would dilute the focus on the interpretation of the results.

7. It is also recommended to cite any reports referring to the safe use of microparticle components especially highlighting the dosage.

Response: We appreciate the recommendation, and note that this is already mentioned in the introduction.

8. English needs to be thoroughly checked and can be improved at many places in the manuscript.

Response: Thank you very much for the comment, the article was completely revised.

Reviewer #3: COMMENTS TO AUTHOR

1. Authors have to give the clarity regarding type of route of administration.

Response: Thank you very much for the comment, it is clarified in line 103 of the version with corrections.

2. Authors have to provide stability data of microparticles as a supplement data.

Response: We welcome the suggestion, which is interesting. However, for the experimental design we did not consider the stability a in storage, but in production as specified in the manuscript by means of the zeta-potential. Storage stability of the optimal formulation under different conditions, as well as throughout the gastrointestinal tract is a study in progress.

3. what is the charge of PEITC? Justify its effect on stability and entrapment.

Response: The PEITC charge is 0, it is a hydrophobic non-ionic compound, as mentioned in the Introduction, and therefore it is well encapsulated, as well as stabilised with olive oil. For clarity, a comment was added on line 497 of the version with corrections.

4. Author have to justify use of lower level of XA variable. Why 0 value as a lower level. If polymer concentration is zero than how microparticles will form?

Response: XA is not the amount of polymer, it is the olive oil/polymer ratio, that is, what is 0 is the olive oil in the formulation with XA at its lowest level.

5. Justify use of olive oil in formulations.

Response: Thank you very much for the comment, that is specified in the introduction.

---

## [Decision Letter · Decision Letter 1]

26 Apr 2021

Chitosan-olive oil microparticles for phenylethyl isothiocyanate delivery: optimal formulation

PONE-D-21-05688R1

Dear Dr. Coscueta,

We’re pleased to inform you that your manuscript has been judged scientifically suitable for publication and will be formally accepted for publication once it meets all outstanding technical requirements.

Kind regards,

Vivek Gupta

Academic Editor

PLOS ONE

Additional Editor Comments (optional):

Reviewers' comments:

Reviewer's Responses to Questions

**Comments to the Author**

1. If the authors have adequately addressed your comments raised in a previous round of review and you feel that this manuscript is now acceptable for publication, you may indicate that here to bypass the “Comments to the Author” section, enter your conflict of interest statement in the “Confidential to Editor” section, and submit your "Accept" recommendation.

Reviewer #2: All comments have been addressed

Reviewer #3: All comments have been addressed

2. Is the manuscript technically sound, and do the data support the conclusions?

Reviewer #2: Yes

Reviewer #3: Yes

3. Has the statistical analysis been performed appropriately and rigorously? 

Reviewer #2: Yes

Reviewer #3: Yes

4. Have the authors made all data underlying the findings in their manuscript fully available?

Reviewer #2: Yes

Reviewer #3: Yes

5. Is the manuscript presented in an intelligible fashion and written in standard English?

Reviewer #2: Yes

Reviewer #3: Yes

6. Review Comments to the Author

Reviewer #2: (No Response)

Reviewer #3: Author has completed revision as per comments. All comments adress carefully so I recommended article for publication.

7. PLOS authors have the option to publish the peer review history of their article (what does this mean?). If published, this will include your full peer review and any attached files.

Reviewer #2: **Yes: **Vineela Parvathaneni

Reviewer #3: No

---

## [Editor Report · Acceptance letter]

28 Apr 2021

PONE-D-21-05688R1 

Chitosan-olive oil microparticles for phenylethyl isothiocyanate delivery: optimal formulation 

Dear Dr. Coscueta:

I'm pleased to inform you that your manuscript has been deemed suitable for publication in PLOS ONE. Congratulations! Your manuscript is now with our production department. 

Kind regards, 

on behalf of

Dr. Vivek Gupta 

Academic Editor

PLOS ONE